



# Source-load matching and energy storage optimization strategies for regional wind-solar energy systems

Yongqing Zhu[*], Qingsheng Li, Zhen Li, Zhaofeng Zhang

Power Grid Planning and Research Center of Guizhou Power Grid Co., Ltd, Guizhou, China

*Correspondence to:* Yongqing Zhu (zzz660880@163.com)

**Abstracts:** In response to the issue of limited new energy output leading to poor smoothing effects on grid-connected load fluctuations, this paper proposes a load power smoothing method based on "one source multiple loads." The method comprehensively considers the proximity of the source and load, as well as the correlation between their power fluctuations, using this as a tracking evaluation standard for source-side and different load-side matching in regional power grids. Initially, loads are clustered and divided based on power frequency division. The EEMD algorithm is then applied to obtain wind and solar energy outputs with greater complementarity and smoother fluctuations, leveraging their low-frequency correlation. Subsequently, a load tracking coefficient is used to compare the matching degree between wind-solar power output and different loads, selecting the most compatible load and output for source-load matching and smoothing. Concurrently, a gray wolf optimization algorithm based on Tent-chaotic mapping is employed to optimize edge energy storage at different load sides, minimizing overall grid-connected load power fluctuations. Numerical results demonstrate that the proposed method can fully utilize the stable output from the low-frequency correlation of wind and solar energy, combined with energy storage, to significantly reduce the fluctuation rate of regional grid-connected loads. This effectively promotes local absorption of source loads, thereby alleviating the pressure on the grid side caused by the randomness and volatility on both sides of the source load.

**Keywords:** Load Power smoothing, Source-Load matching, EEMD Algorithm, grid stability, grid stabilization strategy

## 1. Introduction

In response to China's dual carbon goals, new power systems utilizing renewable energy sources like wind and photovoltaic are rapidly advancing. The installed capacity of wind turbines and photovoltaic units, crucial components of renewable energy, is growing (Xi, 2020; Gao, 2022). However, both wind and photovoltaic power generation are highly volatile and stochastic, leading to increased pressure on grid-side dispatch when parallelized with traditional load demands (Qu and Ye, 2023; Lee and Baldick, 2017; Ma et al., 2020; Oh and Son, 2022; Li et al., 2022). Often, the installed capacity of wind-solar units in a region is insufficient to meet local load demands, or their utilization is limited, resulting in low efficiency in suppressing load fluctuations.

Despite these challenges, the consistency of regional source-load fluctuations can be leveraged to improve local consumption



of wind-solar power and reduce grid-side pressure from load power fluctuations, which is crucial for regional grid-connected dispatch. One effective strategy is the use of wind-solar correlation for regional power suppression, which has been extensively studied (Liang et al., 2023; Hu et al., 2024; Xie et al., 2017; Tan et al., 2022; Dong et al., 2018; Haensch et al., 2024; Wang et al., 2020; Zhao et al., 2020). By considering the complementary characteristics of wind and solar power, volatility and randomness in original output can be reduced. For instance, typical wind-solar output scenarios can be generated based on wind-solar correlation, aiding in optimal scheduling for microgrids.

The traditional energy optimization dispatching strategy is distinct from the source-load matching strategy, which focuses on regional renewable energy consumption and grid-connected power fluctuation reduction. Source-load matching is implemented based on evaluating the load tracking degree, which considers the smoothness of the load tracking and residual load curve (Zhu et al., 2024). To enhance load tracking, different tracking coefficient models are established based on the overall system fluctuation's smoothness (Shi et al., 2023; Mitrofanov and Baykasenov, 2022; Beluco et al., 2008). Additionally, the Copula function can evaluate source-load matching, inverting the energy side's complementary characteristics (Ren et al., 2024). However, these methods are often limited to considering power differences or fluctuation similarities between the source and load, or they only address matching between single power and load sides.

In this paper, we propose a source-load matching strategy based on wind-solar complementarity and the "one source, multiple loads" concept. We prioritize the more stable low-frequency output of wind-solar to match load power fluctuations according to load tracking criteria. We also optimize the edge storage charging and discharging strategy for each load group using the gray wolf optimization algorithm with Tent-chaotic mapping, aiming to minimize overall load fluctuation in regional grid connections and reduce power fluctuations on both sides of the grid.

Unlike current research on microgrid or regional source-load matching models, which typically consider a single power side and a single load group, this paper delves deeper into the impact of different power-side suppression abilities on various load groups, influencing regional grid fluctuations. We construct a "one source, multiple load" regional grid framework, utilizing a typical wind-solar co-generation plant and multiple load groups with edge storage. K-medoid clustering is used to categorize loads into groups with typical energy use characteristics. Based on the complementary low-frequency correlation of wind-solar power, the source-side power output is smoothed. The proposed load tracking index is then employed to track load-side power fluctuations, reducing regional grid-connected power fluctuations.

The framework of "the one source, many loads" regional grid is shown in Fig.1.





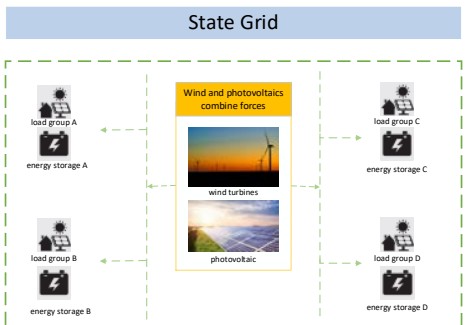

**Fig.1 "One source with multi-load" regional power grid framework**

The specific steps of power leveling are shown in Fig.2.

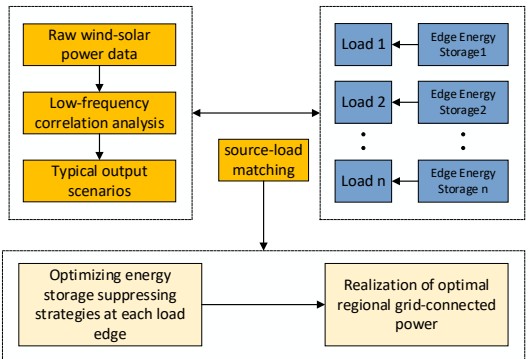

**Fig.2 Flow chart of regional source-load matching and stabilizing method**

The primary contribution of this paper is as follows:

• Frequency decomposition of the daily wind-solar output, correlation analysis of the decomposed low-frequency components and obtaining a typical daily scenario set of wind-solar low-frequency output, use Euclidean distance to judge the scenario set scenario and the original output of the corresponding day, and select the closest day as the replacement of the output of that day.

• The load is clustered based on the rough K-means of variational firefly optimization. The load tracking evaluation criteria proposed in this paper are used to compare the matching degree between the output scenario and each load group. The load group with the highest source-load matching degree is selected as the output satisfaction object for that day.

• The gray wolf optimization algorithm based on Tent-chaotic mapping is used to optimize each load-side edge energy storage leveling strategy to minimize the fluctuation of regional grid-connected load, promote the level of wind-solar consumption, and reduce the pressure of grid-side dispatch.



## 2. Source-Load Matching for Regional Wind-solar Systems

### 2.1 CEEMD-Based Wind-Solar Output Frequency Decomposition

In this paper, based on the previous study (Mahdavi et al., 2023), we further study the smoothing effect of the source-side power output on the load-side fluctuation. Based on not changing the capacity configuration in the original region, the obtained typical daily scenario set of wind-solar power output and the load scenario is source-load matched to achieve the power fluctuation smoothing on the regional grid connection.

To better achieve the decomposition effect, this paper adopts the CEEMD (complementary ensemble empirical mode decomposition, CEEMD) algorithm to decompose the frequency of the original wind-solar output data. CEEMD has the characteristics of independent homogeneous distribution and opposite sign for the white noise added to the original signal for the auxiliary decomposition. It can compensate for the shortcomings of modal mixing in the traditional empirical mode decomposition (EMD) while better reducing the noise remaining in the original signal and decomposition errors (Liu et al., 2024). The specific decomposition steps are as follows:

1)      Add a positive and negative pair of random Gaussian white noise, respectively, to the original sequence under:

$$X^+(t) = x(t) + \mu^+(t) \tag{1}$$

$$X^-(t) = x(t) + \mu^-(t) \tag{2}$$

where $X^+(t)$, $X^-(t)$ are the sequences after adding positive and negative random Gaussian white noise $\mu^+(t)$, $\mu^-(t)$ respectively.

2)      EMD decomposition of the newly generated signal to obtain the inherent modal functions (IMF) components of each order:

$$X^+(t) = \sum_{i=1}^{m} c_i^+(t) + r^+(t) \tag{3}$$

$$X^-(t) = \sum_{i=1}^{m} c_i^-(t) + r^-(t) \tag{4}$$

where $c_i^+(t)$, $c_i^-(t)$ are the i-th IMF component of the decomposition, $r^+(t)$, $r^-(t)$ are the remaining terms of the decomposition.

3)      Repeat the above steps n times. Each repetition adds a new and different sequence of paired Gaussian white noise.

4)      Summing the IMF components obtained from each repetition to take the mean value as the final decomposition result. The final $c_i(t)$ and $r(t)$ are expressed as follows:



$$c_i(t) = \frac{1}{2n} \sum_{i=1}^{n} (c_{ji}^+(t) + c_{ji}^-(t))$$

101     (5)

$$r(t) = \frac{1}{n} \sum_{i=1}^{n} (r_j^+(t) + r_j^-(t))$$

102     (6)

where $c_{ji}^+(t)$, $c_{ji}^-(t)$ are the i-th IMF component obtained from the decomposition at the j-th repetition; $r_j^+(t)$, $r_j^-(t)$ are the
residuals obtained from the decomposition at the j-th repetition; $c_i(t)$ is the i-th IMF component from the final
decomposition; $r(t)$ is the remaining amount from the final decomposition.

### 2.2 Rough Load Clustering Optimized By Mutation Firefly Algorithm

This paper uses a variational strategy and a firefly algorithm with differential evolution to optimize the traditional clustering
algorithm (Wei et al., 2023). The rough K-means algorithm is an improvement of the classical K-means algorithm, and the
difference is that the algorithm divides the sample objects that cannot be determined into the boundary set of the class. The
division is based on the presence or absence of other clustering centers with a difference between the distance and the
minimum distance from the sample object less than a given threshold.
The core concepts of rough set theory are upper approximation and lower approximation rather than boundary domain, and
the variation of the number of objects in the lower approximation and boundary set and the variability of object distribution
dynamically adjust the center-of-mass weights. The relative distances are:

$$T' = \left\{ t : \frac{d(x_n, m_k)}{d(x_n, m_h)} \leq \xi \wedge h \neq k \right\}$$

(7)

The variant firefly optimization algorithm makes full use of the information of individual firefly populations through a
double variant strategy, which significantly improves the ability of the algorithm to jump out of the local optimum and to
converge to the global optimum with probability one under a large enough number of iterations. The new objective function
value is constructed as the firefly light intensity for the initial clustering centroid search, and the optimal solution found by
the firefly algorithm is used as the clustering center of the algorithm for clustering iterations:

$$I = f(x) = \left( \frac{O}{I} \right)^{\lambda}$$

(8)

where I is the intra-class distance, which is the sum of the distances from each data sample in each class to its cluster center;
O is the inter-class distance, which is the distance between the cluster centers; when $\lambda \geq 1$, the data may be out of range if
the number of samples and the number of dimensional bases are large, and $\lambda = 1/2$ is taken in this study.





## 2.3 Load Tracking Evaluation Criteria

This paper considers the proximity of source-load power magnitude and the correlation degree of source-load power fluctuation as the evaluation criteria of source-side load tracking. Based on the fact that Spearman's coefficient and Euclidean distance present complementary advantages and disadvantages in measuring correlation, Euclidean distance, and rank correlation coefficient are used to calculate them, respectively. The following equation is shown:

$$\max \theta^i = \alpha_1 \delta_1^i + \alpha_2 \delta_2^i \tag{9}$$

where $\theta^i$ is the match between the source-side output and the i-th load group; $\delta_1^i$ is the tracking coefficient between the source-side output and the i-th load group; $\delta_2^i$ is the correlation between the normalized source-side output and the i-th load group; $\alpha_1$ and $\alpha_2$ are the weight coefficients of the corresponding indicators, and the initial ratio of the two is selected as 1:1 in this paper, considering their different effects on the matching degree.

$$\lambda^i = \sqrt{\sum_{t=1}^{T} (P^t - L_i^t)^2} \tag{10}$$

$$\xi_1^i = \frac{\lambda^i}{\sum_{n=1}^{N} \lambda^n} \tag{11}$$

where $P^t$ and $L_i^t$ are the output power and load power of the i-th load group at moment t, respectively, T is the number of moments of that day (T=24); $\lambda^i$ and $\lambda^n$ are the source-side output and the i-th and n-th load group power Euclidean distances, respectively, for that day, and N is the number of load groups; $\xi_1^i$ is the power Euclidean distance between the normalized source-side output and the i-th load group:

$$\delta_1^i = 1 - \xi_1^i \tag{12}$$

Spearman rank correlation coefficient was used to do a correlation analysis between wind-solar low-frequency output and each load power. Spearman correlation coefficient is a non-parametric statistical method of rank correlation using monotonic equations in statistics to evaluate the correlation between two statistical variables. The basic idea is that there are three binary distributions of random vectors $(m_1, n_1), (m_2, n_2), (m_3, n_3)$ with the difference between the probability that at least one of them occurs in concert with the other distributions and the probability that at least one of them does not occur in concert with the other distributions as the correlation indicator describing the random variables (Wei et al., 2023), which is calculated as the following equations:

$$\tau = 1 - [6\sum_{t=1}^{T} d_t^2 / T(T^2 - 1)] \tag{13}$$





where $\tau$ is the Spearman correlation coefficient between any two vectors; T is the vector dimension, which in this paper is the 24 time periods that divide each day in source-load matching; d is the set of element ranking differences in the two vectors.

$$\delta_2^i = \frac{\tau^i}{\sum\limits_{n=1}^{N} \tau^n} \tag{14}$$

To make a uniform distance and correlation variation relationship, where $\tau^i$ and $\tau^n$ are the correlation coefficients between the source-side output and the i-th and n-th load groups, respectively, for that day, and N is the number of load groups.

According to the above load tracking evaluation criteria, the matching degree between the wind-solar system's low-frequency output and each load's power is compared, and the most matching load is selected as the target of the power leveling on that day. Among them, the wind-solar excess energy is used to charge the energy storage corresponding to the matched load. When the load is not matched with the energy output day, if the load has too much fluctuation, the energy storage according to its own SOC state and the set fluctuation threshold, the load is smoothed to a certain extent; the wind-solar output has excess energy in a specific period, the energy is used to charge the energy storage, so that on the day when the load is not matched, the energy storage has a specific scheduling interval, using renewable resources, reducing the pressure on the grid, and realizing It can be used to calm down the fluctuation of load and avoid the waste of energy that may exist when the wind-solar power is connected to the grid (Luo et al., 2021).

## 3. Load Edge Energy Storage Suppressing Strategy

In order to better achieve the overall grid-connected power fluctuation smoothing of regional loads, the charging and discharging strategies of small-capacity energy storage on each load group side are optimized by using the Gray Wolf algorithm based on Tent chaotic mapping to minimize the overall fluctuation rate of regional loads. The method proposed in this paper, compared with the traditional energy storage method, can optimize the single-period load reduction to a more detailed multi-time period reduction, avoiding the dispatch pressure on the grid after a substantial load smoothing after the load rises again during peak and valley periods, to achieve the reduction of the overall fluctuation rate.

### 3.1 Gray Wolf Optimization Algorithm Based on Tent-chaotic Mapping

Compared with the traditional particle swarm algorithm and genetic algorithm, the gray wolf algorithm has a good performance in terms of the accuracy of solving the problem and the convergence speed due to its strong convergence performance, simple structure, few parameters to be adjusted, and the ability to achieve a balance between local optimization and global search (Wei et al., 2023). This paper uses the improved Gray Wolf optimization algorithm with Tent chaotic mapping to flatten the marginal energy storage on different load sides.



The core idea of the Gray Wolf Algorithm is to mathematically model the social hierarchy of gray wolves in GWO by
defining the top 3 best wolves (optimal solutions) as $\alpha$, $\beta$, $\delta$ each, which guides the other wolves in their search toward the
goal. The remaining wolves (candidate solutions) are defined as $\omega$, and they update their position around $\alpha$, $\beta$, $\delta$.
Chaos has randomness and traversal and initial value sensitivity to speed up the convergence of the algorithm, generating
chaotic sequences based on Tent mapping to initialize the population:
$$Z_{I+1}^{k} = \begin{cases} \dfrac{Z_{I}^{k}}{u}, 0 \leq Z_{I}^{k} \leq u \\ \dfrac{1-Z_{I}^{k}}{1-u}, u < Z_{I}^{k} \leq 1 \end{cases} \tag{15}$$

where, k is the number of populations, I is the number of current iterations, and to maintain the randomness of the
initialization information of the algorithm, u takes the value of $u \subset rand(0,1)$. Combined with the chaotic sequence $Z_{I}^{k}$,
the further process of generating the sequence $X_{I}^{k}$ of initial locations of individual gray wolves in the search area is as
follows:
$$X_{I}^{k} = X_{I,\min}^{k} + Z_{I}^{k}\left(X_{I,\max}^{k} - X_{I,\min}^{k}\right) \tag{16}$$

where, $X_{I,max}^{k}$, $X_{I,min}^{k}$ is the maximum and minimum value of $X_{I}^{k}$, respectively.
A dynamic weight factor b, which changes in a linearly decreasing manner, is introduced to update the gray wolf individual
step size dynamically:
$$b(I) = b_f - \frac{I}{MaxIter}\left(b_f - b_s\right) \tag{17}$$

where, $b_s$, $b_f$ denotes the initial and final values of the weighting factors, respectively.
A fitness scaling factor was introduced to dynamically weight the averages to differentiate head wolf contributions, thus
effectively differentiating the different guiding roles of head wolf $\alpha$, $\beta$, $\delta$ on subsequent gray wolf individual position
updates:
$$\begin{cases} f = \left|f_\alpha + f_\beta + f_\delta\right| \\ v_1 = \dfrac{f_\alpha}{f}, v_2 = \dfrac{f_\beta}{f}, \quad v_3 = \dfrac{f_\delta}{f}, f > 0 \\ v_1 = v_2 = v_3 = \dfrac{1}{3}, f = 0 \end{cases} \tag{18}$$

where, $v_1$, $v_2$, $v_3$ is the adaptation scale factor; $f_\alpha$, $f_\beta$, $f_\delta$ is the adaptation value of $\alpha$, $\beta$, $\delta$ respectively.
The fused improved position update formula is:




$$X(I+1) = b(I) \cdot r_4 \cdot$$
$$(v_1 \cdot X_1 + v_2 \cdot X_2 + v_3 \cdot X_3)$$
(19)

where, $r_4$ is a random vector between [0,1].

## 3.2 Edge Energy Storage Optimization Model

The gray wolf optimization algorithm based on Tent-chaotic mapping is used to optimize the charging and discharging power
of the edge energy storage of the remaining load groups with the objective of to minimize the fluctuation of the regional
required grid leveling load to achieve the reduction of the regionally grid-connected load fluctuation. The optimization
objective function is as follows:
$$\min F_i = (\sum_{t=1}^{T} \frac{M_i(t+1) - M_i(t)}{M_i^{\max}}) / 24$$
(20)

where, $F_i$ is the overall regional load fluctuation rate on day i; $M_i$ is the overall regional load power on day i after
source-load matching; $M_i^{max}$ is the maximum load value on that day; and T is the number of moments on that day (T=24).
The constraints are:
1)        Wind farm operation constraint
$$0 \le P_{wind,s,t} \le P_{wind,s,t}^{\max}$$
(21)

2)        Photovoltaic plant operation constraint
$$0 \le P_{PV,s,t} \le P_{PV,s,t}^{\max}$$
(22)

3)        Load constraint
$$P_{LD,s,t}^{\min} \le P_{LD,s,t} \le P_{LD,s,t}^{\max}$$
(23)

4)        Energy storage constraint
Energy storage charging and discharging power constraint:
$$P_{ess,s,t}^{\min} \le P_{ess,s,t} \le P_{ess,s,t}^{\max}$$
(24)

Energy storage charge state constraint:
$$SOC^{\min} \le SOC_{s,t} \le SOC^{\max}$$
(25)

Energy storage discharge balance constraint:





$$\sum_{t=1}^{T} P_{ess}(t) = 0 \tag{26}$$

### 3.3 Energy Storage SOC Control

The basic idea of energy storage leveling is: on the day when the load matches the energy source, if the load is larger than
the output and fluctuates widely, energy storage discharges to level the load, and if the load is smaller than the output, energy
storage charges to avoid the waste of renewable energy; at the same time, on the day when the load does not match the
energy output, if the load has a significant fluctuation, energy storage, according to its own SOC state and the set fluctuation
threshold, will level the load to a certain extent. To better protect the energy storage and prolong the life of the storage.
In order to better protect the energy storage and prolong the life of the energy storage, it is necessary to limit the energy
storage ground charge and discharge, i.e., the energy storage SOC state is limited to [0.1, 0.9]. The SOC is calculated as
follows:
Discharge:
$$S_{soc}(t) = (1-\rho)S_{soc}(t-1) - \frac{P_e(t)\Delta t}{E\eta_d} \tag{27}$$
Charging:
$$S_{soc}(t) = (1-\rho)S_{soc}(t-1) - \frac{P_e(t)\Delta t\eta_c}{E} \tag{28}$$
where, $S_{soc}(t)$ and $S_{soc}(t-1)$ denote the SOC values of energy storage in period t and t-1, respectively; $P_e(t)$ denotes the
required leveling target of energy storage in period t; $\Delta t$ is the length of period; $\rho$ is the self-discharge rate; $\eta_d$ and $\eta_c$
denote the energy storage discharge efficiency and charging efficiency, respectively; and E is the energy storage capacity.

### 4. Experiments and Results

This paper analyzes the actual output power of a 100MW wind farm and a 50MW PV co-generation farm and the actual
loads of four typical load groups in the region in the summer of 2018 in a northwestern area.
From the scenario generation method described in the previous section, a typical scenario of wind and solar low-frequency
output power is obtained, as shown in Fig.3.
They are adopting the load-tracking evaluation criteria proposed in Section 2.3. Furthermore, combined with the local
weather, the daily corresponding wind power and different load groups are matched and evaluated, and the load group with
the highest degree of matching is selected as the main suppression target of the wind power on that day.





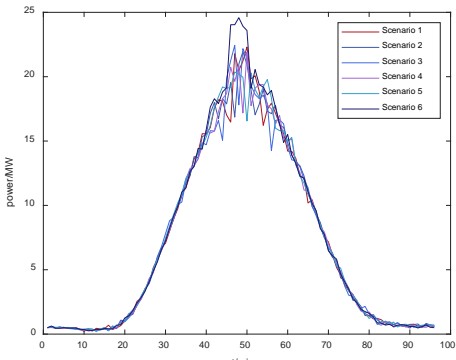
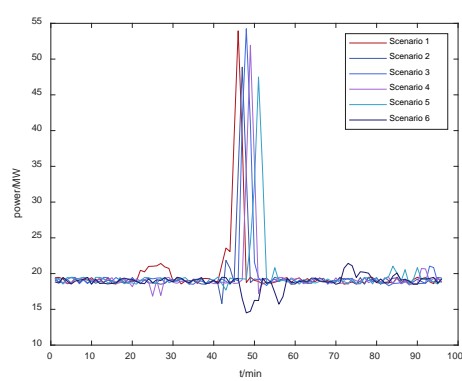


(a) Generation results of wind power scenario     (b) Generation results of PV scenario
**Fig.3 Scenery scene generation results**
Table 1 shows the matching degree between source-side output and different load clusters and the original load for a
particular day, where load cluster 5 is the original load before the clustering of loads. The table shows that the matching
degree between source-side output and original load is less than 0.3, while the highest matching degree of the clustered load
groups can reach 0.52. Therefore, this paper can effectively explore the matching degree between source-side output and
typical load groups after dividing the load clusters.

**Table 1 Comparison of matching degrees on a certain day**

| Load group | Matching degree |
| --- | --- |
| Load group 1 | 0.39 |
| Load group 2 | 0.31 |
| Load group 3 | 0.52 |
| Load group 4 | 0.24 |
| initial load | 0.28 |


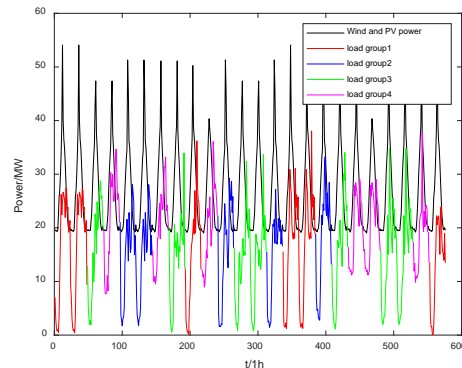


**Fig.4 Source-load matching results**
According to the method described in the previous paper, the matching results are shown in Fig.6, and the wind-solar output



is based on the principle of the highest matching degree to meet different loads daily. As shown in the figure, the
load-tracking evaluation criteria established in this paper can select the load with the most closely matched output among
different loads for matching, reducing the grid-side pressure on both sides of independent dispatch. At the same time, the
load side is split into different load groups. The wind-solar output has excess energy at a specific period, which is used to
charge the energy storage so that the energy storage has a specific dispatch interval on the days when the load is not matched.
The suppressing time of the energy storage can be further extended.

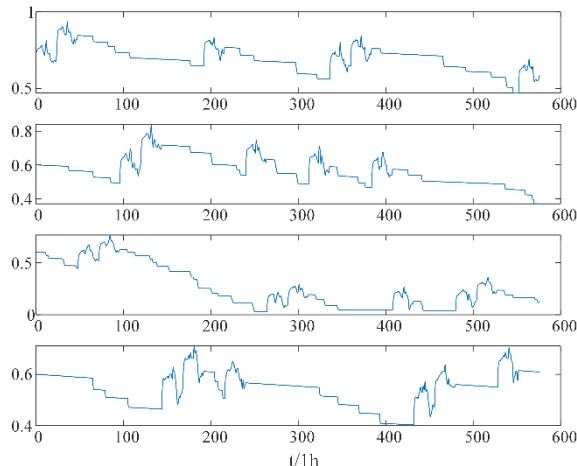


**Fig.5 Energy storage SOC state**
As shown in Fig.5, the SOC state of each load-side edge energy storage after optimizing the overall load fluctuation in the
region using the gray wolf optimization algorithm. The figure shows that the source-load matching can provide enough
energy for the energy storage to meet its required smoothing objective, and the SOC of each energy storage is maintained in
the ideal interval to avoid damage to the energy storage lifetime. In this paper, the selected energy storage parameters are
shown in Table 2.

**Table 2 Energy storage system parameters**

| Parameter type | Storage batterie |
|---|---|
| Maximum continuous discharging power /MW | 10 |
| Maximum continuous charging power /MW | 10 |
| Rated capacity /MW· h | 5 |
| Permissible depth of discharge /% | 10~90 |
| The initial state of charge /% | 60 |
| Self-discharge rate /(%/h) | 0.6 |
| Charge and discharge efficiency /% | 95 |






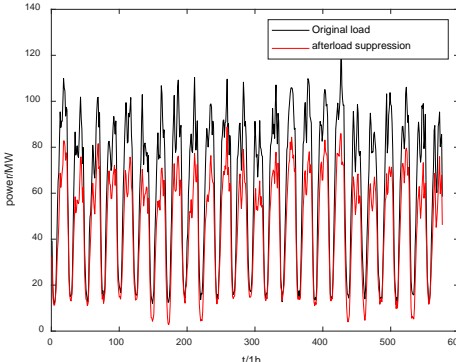

**Fig.6 Load power changes before and after the suppression**

As shown in Fig.5, the overall load power of the region is compared with the original regional load power after adopting the method proposed in this paper; the source-load matching strategy proposed in this paper can significantly reduce the power target of the grid-side load to be leveled, and reduce the pressure of the grid-side to meet the original load. At the same time, the method in this paper makes reasonable use of the regional wind-solar power and load adjacent to the characteristics of easy scheduling, the use of source-load matching strategy to achieve the power of local consumption, used to suppress the fluctuations in the load at the same time to avoid the wind-solar power in the grid-connected energy waste situation that may exist.

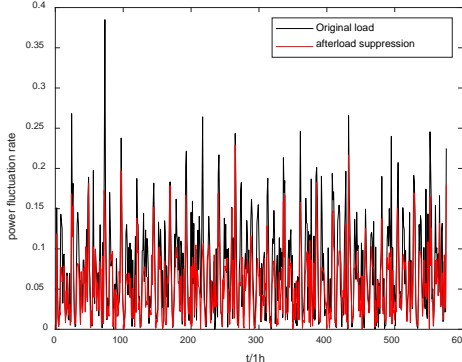

**Fig.7 Changes in load volatility before and after smoothing**

As shown in Fig.7, the overall regional load fluctuation rate is compared with the original regional load fluctuation rate after adopting the proposed method in this paper. As shown in the figure, the proposed method can significantly reduce the fluctuation in the original regional load. The fluctuation of the original load can reach about 0.4, which is a tremendous pressure on the grid dispatch. However, after adopting the proposed method, the fluctuation rate of the regional load is reduced to less than 0.2, which reduces the difficulty of grid-side dispatch.

To further verify the effect of the proposed method on regional load fluctuation, three scenarios are set up in this paper for




comparison. Scenario 1 is the traditional regional load suppression, i.e., the load power is all satisfied by the grid side;
Scenario 2 is the wind-solar system low-frequency output power used to satisfy the load power, while the energy storage
suppresses a certain amount of excess wind-solar output and load fluctuation; Scenario 3 is the proposed method.

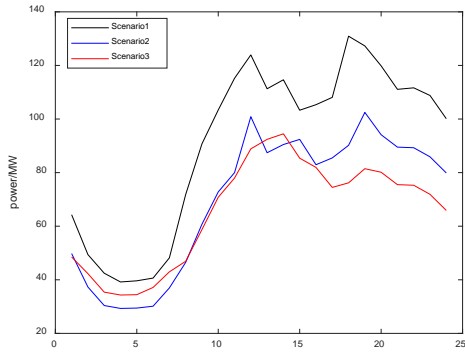


**Fig.8 Comparison of Scenario 2 and Scenario 3**
As shown in Fig.8, the comparison between Scenario II and Scenario III is shown. As shown in the figure, compared with
the direct use of wind-solar power to meet the load, the method proposed in this paper is more effective in suppressing the
peak fluctuation of the load and reducing the load fluctuation rate. As scenario 2 is the direct suppression of the
low-frequency output of wind-solar power, the degree of load reduction in scenario 2 is higher than that in scenario 3 at the
peak of the wind-solar power, which to some extent aggravates the pressure on the grid when the load rises at the next
moment. The method proposed in this paper optimizes the charging and discharging of each energy storage to minimize the
overall fluctuation of the regional load when reducing the load power during the peak hours, charging the energy storage
appropriately during the load valley section, and avoiding the fluctuation caused by over satisfying the low valley load.


**Fig.9 Power comparison in different scenarios on a day**
A comparison of the overall load power on the region for three scenarios on a randomly selected day is shown in Fig.9. As





shown in the figure, scenario three's overall fluctuation rate is smaller than scenario two's. The method proposed in this paper can provide overall smoothing of the split load while the remaining energy from the source-load matching is stored in the energy storage so that the load can be smoothed to some extent even when it is not matched. Compared with Scenario 2, Scenario 3 has a higher load power in part of the time, which is because the objective of the proposed method is to reduce the overall volatility of the load, so part of the wind-solar power is used to charge the energy storage in that time. Compared with the traditional wind-solar power directly used to meet the load, the method proposed in this paper can divide the load reduction of a single period into the reduction of multiple periods and realize the lowest fluctuation of the regional load as a whole.

In order to verify the effectiveness of this paper's method for intraday scheduling, this paper forecasts the load with a time scale of 1 hour. It uses this paper's method for smoothing verification.

As shown in Fig.10, the results of using LSTM to forecast each load based on historical data show that LSTM can forecast the load effectively. In operation scheduling, the next day's load can be predicted based on historical data. At the same time, the source-side output scenario is selected based on the weather, and the source-load matching strategy proposed in this paper is used to match the suppression. In the following, the forecast result of a particular day is selected to analyze the leveling.

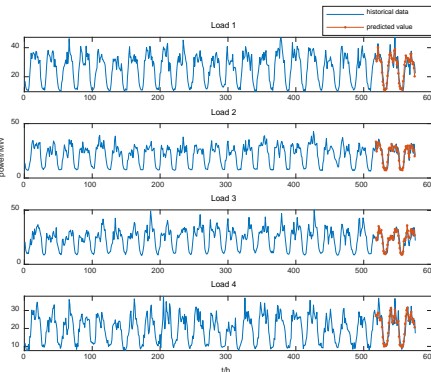

**Fig.10 Load prediction results based on LSTM**

As shown in Fig.11, after forecasting the load on a particular day, the wind-solar power is selected to carry out source-load matching suppression, and the results of the grid-connected load power in the region are compared after the wind-solar power, and the marginal energy storage is suppressed for each load. After using the method in this paper, the overall grid-connected power of the regional load is significantly reduced. At the same time, the peak fluctuation of the load is also significantly reduced, such as between 12:00-14:00 and 16:00-20:00, the original grid-connected load there is a significant peak, there is a certain amount of pressure on the grid scheduling, and the fluctuation after the suppression of the fluctuation of the grid to reduce the negative impact of grid-connected.





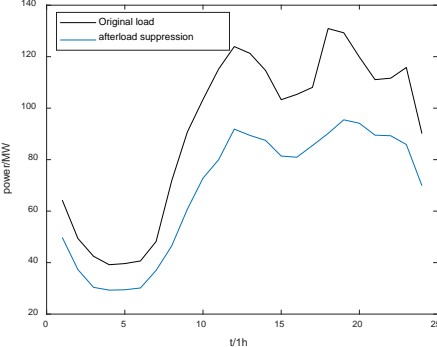

**Fig.11 Predictive flattening results of the regional power grid on a certain day**

As shown in Fig.12 is the change of SOC state of each edge energy storage after the leveling off of the forecast day; as can be shown in the figure, the matching object of the wind-solar power output on that day is the fourth load group, and after the wind-solar power output meets the load demand, the excess energy is used to charge the energy storage, so that the edge energy storage of the matched load can be kept in a good state at the end of the day. At the same time, it can be seen from the change in the SOC state of other marginal energy storage that on an unmatched day, the marginal energy storage corresponding to the load group is appropriately discharged at the peak value of load fluctuation to reduce the load fluctuation. At the same time, to avoid the load group failing to be the matched object of wind-solar power for many consecutive days, the energy storage does not release all of the stored energy at one time so that the leveling-off time of the storage is prolonged as much as possible. The utilization of the storage is improved. The energy storage will not release all of its stored energy at once to extend the leveling time and improve the utilization of energy storage as much as possible.

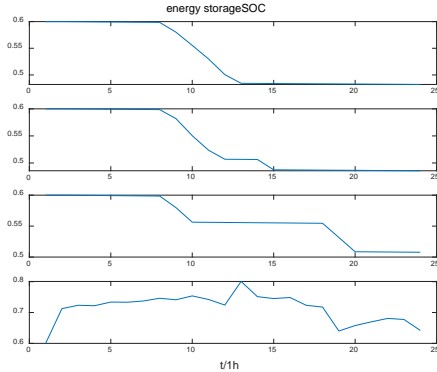

**Fig.12 Edge energy storage day SOC status**

Fig.13 compares grid-connected volatility before and after load suppressing of the regional grid on the forecast day; as demonstrated in the figure, the volatility of the grid-connected load after using the method of this paper is significantly reduced, avoiding the peak value of fluctuations. The grid-connected volatility of the original load has reached 0.2 many





times. In contrast, the volatility after suppressing is maintained at 0.1 or below, which verifies the effectiveness of the method of this paper for grid-connected load suppressing.

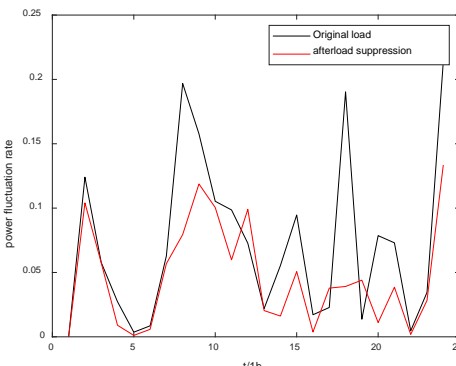

**Fig.13 Comparison of forecast daily volatility before and after flattening**

## 5. Conclusion

This paper addresses the shortcomings of wind-solar power output in the region for load suppressing. We also consider the smoother output wind-solar power low-frequency output and source-load matching strategy for regional load smoothing. The proposed method has several significant features and contributions:

(1) Framework Development**: A regional grid framework of "one source and multiple loads" is proposed. This framework effectively utilizes the low-frequency output of wind-solar power, which is more stable, to match and smooth the load fluctuations. By dividing the load into multiple groups and matching them with the source-side output, the method reduces the overall load fluctuation and the pressure on the grid-side dispatch.

(2) Optimization Algorithm: The gray wolf optimization algorithm based on Tent-chaotic mapping is introduced. This algorithm enhances the global and local optimization capabilities, ensuring that the edge energy storage at each load side is optimized to minimize the overall load fluctuation. The algorithm's chaotic mapping feature helps in avoiding local optima and achieving a more robust solution.

(3) Local Consumption and Grid Pressure Reduction: The method effectively promotes the local consumption of wind-solar power and reduces the pressure on grid-side dispatch. By matching the source and load, the method ensures that the renewable energy is utilized more efficiently, reducing the need for grid support and improving the overall stability of the regional power system.

(4) High Complementarity Utilization: The method fully utilizes the high complementarity of wind and solar power in the low-frequency band. This complementarity helps in reducing the uncertainty and volatility of renewable energy sources, making the power output more predictable and manageable.

(5) Volatility Reduction: The method significantly reduces the volatility of the regional power grid. By optimizing the



charging and discharging strategies of edge energy storage, the method ensures that the load fluctuations are minimized, reducing the difficulty of grid-side dispatch and improving the reliability of the power system.

In summary, the proposed method provides a comprehensive solution to the challenges of integrating renewable energy into the grid. It not only improves the efficiency of renewable energy utilization but also enhances the stability and reliability of the power system. The method's ability to match the source and load effectively and optimize energy storage operations makes it a valuable tool for regional grid management. Future work will focus on further refining the model and exploring its application in different regional and operational contexts to maximize its potential in promoting sustainable energy use and grid stability.

**Code/Data availability**

Not applicable.

**Competing interests**

The authors declare that they have no conflict of interest.

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
