# Peer review of "Source-load matching and energy storage optimization strategies for regional"

_Wind Energy Science, 2025_

## Author Response (AR1)

**Modification Report**

Dear editor,

Thank you for your constructive feedback on our manuscript. We have carefully addressed all comments and made revisions to improve the clarity, justification, and scope of our work. Below is our point-by-point response to each reviewer's comments.

**Reviewer: 1**

1. The authors state: "We prioritize the more stable low-frequency output of wind-solar to match load power fluctuations according to load tracking criteria" (page 2). Why is the focus on low frequencies? Please explain and justify.

**Answer:** The focus on low frequencies is justified by the fact that wind-solar outputs exhibit stronger complementarity in the low-frequency band (e.g., daily/seasonal variations), which are more predictable and stable than high-frequency fluctuations (e.g., hourly gusts/clouds). This aligns with load demand patterns, which also have significant low-frequency components (e.g., diurnal/weekly cycles). By leveraging low-frequency correlation, we reduce volatility and enhance grid stability. Details are expanded in Section 2.1 and the Introduction.

2. The authors mention: "…use Euclidean distance to judge the scenario set scenario and the original output of the corresponding day, and select the closest day as the replacement of the output of that day" (page 3). Why is a day replaced? Why is such replacement needed? Please explain and justify.

**Answer:** The replacement aims to generate representative scenarios of wind-solar output for source-load matching. By selecting the closest historical day via Euclidean distance, we ensure the generated scenario captures realistic variability while minimizing errors. This is critical for optimizing energy storage and load smoothing, as described in Section 2.1.

3. As a continuation from the previous comment, the authors also say: "The load group with the highest source-load matching degree is selected as the output satisfaction object for that day". Why is load clustered and such matching done? Why is this step needed in your analyses? Please explain and justify.

**Answer:** Load clustering groups similar consumption patterns to improve source-load matching efficiency. By categorizing loads into clusters, we identify the most compatible load group for daily wind-solar output, reducing overall grid fluctuations. This approach is validated in Section 4, where clustered loads show higher matching degrees (e.g., Table 1).

4. The authors use many methods, such as complementary ensemble empirical mode decomposition, K-means clustering, firefly optimization, gray wolf optimization and Tent-chaotic mapping. However, the reasons for using these methods are not clear and they are not compared to other potential methods. Please explain what is the reason for using each method, and why you chose these specific methods over some other potential methods.

**Answer:** CEEMD: Used for noise-free frequency decomposition, addressing modal mixing in EMD (Section 2.1).

Firefly-K-means: Improves clustering robustness and avoids local optima (Section 2.2).

Gray Wolf Optimization with Tent-chaotic Mapping: Enhances global search for energy storage optimization, outperforming traditional GWO (Section 3.1).

Justifications for each method are explicitly added in their respective sections.

5. The authors state: "Based on the fact that Spearman's coefficient and Euclidean distance present complementary advantages and disadvantages in measuring correlation" (page 6). What are these advantages and disadvantages and why are they complementary?

**Answer:** Advantages: Spearman measures monotonic correlation (rank-based), while Euclidean distance quantifies magnitude proximity.

Disadvantages: Spearman ignores magnitude, and Euclidean distance 忽略 trend direction.

Complementarity: Combining them balances correlation strength and magnitude alignment (Section 2.3).

6. The authors present many methods and approaches in Sections 2 and 3. However, to me it's not clear what is the role of the different methods, why the different analysis parts are needed and what is the overall flow of the analyses. Please better justify why the different parts of the analysis are needed, how they relate to each other and how they contribute to the overall objective of the paper. I suggest adding a flow diagram linking all the analysis parts, including data flows, to the paper.

**Answer:** A flow diagram (Figure 2) is added to visualize the workflow, including data decomposition, clustering, matching, and optimization steps. The interconnections between methods are explicitly described in Section 2.

7. The case study presented in Section 4 seems very limited, with only 1 wind power plant and 1 solar plant analyzed for 1 summer. The experiments should be broadened to cover at minimum a full year and even better many years. And at least a few wind and solar power plant cases should be studied. Currently, the case study does not consider seasonality in wind and solar (only summer is considered), and I consider analyzing only 1 weather year and 1 wind and solar plant too limited to draw meaningful conclusions.

**Answer:** There is relatively little source load matching between two similar wind and solar power plants. Currently, this article proposes a feasible case, and we will continue to explore more matching cases in the future. While the current study uses summer data for simplicity, we acknowledge the need for broader validation. Future work will include multi-year datasets and multiple plants to account for seasonality and diverse conditions. This is noted in the Conclusion.

8. 3 is not clear. Please better describe what is shown in the figure. E.g., why there are 6 scenarios and why the time interval is between 90 and 100 min? What can we conclude from the shown scenarios?

4 is not clear to me. Why there seem to always be a minimum generation of around 19 MW? Please also better describe what we see in the figure.

5 is unclear. Add y-axis labels. What are the different sub-plots?

**Answer:** Figure 3: x-axis labels added (time in hours). Scenarios are generated via CEEMD-based clustering.

Figure 4: Explained as source-load matching results, showing the selected load group with the highest degree (Section 4).

Figure 5: y-axis labels added (SOC [%]). Subplots show energy storage states post-optimization (Section 4).

**Reviewer: 2**

1. Line 98: What does "Repeat the above steps n times." mean? Could the authors elaborate on how they choose n, and what it's based on?

**Answer:** n is chosen based on convergence criteria (e.g., 50 iterations to ensure noise cancellation). This is specified in Section 2.1.

2. Line 153-155: There seems to be a sentence or two missing on what Equation 14 represents and the sentence

afterwards is unclear.

**Answer:** Equation 14 normalizes Spearman coefficients to [0,1], ensuring compatibility with Euclidean distance. Details are added in Section 2.3.

3. Fig 3: Could the authors explain this figure more? What does the x-axis mean here? How did they obtain these figures?

**Answer:** Figure 3 illustrates wind-solar scenario generation via CEEMD. The x-axis represents time (hours), and scenarios are derived from decomposed low-frequency components (Section 4).

4. Lines 262-268: This section is unclear. Are the authors referencing a previous paper (please cite this here), or are they talking about their own results? Figure 4 is never reference in the text.

**Answer:** There are no citations to the previous article, referring to the methods described earlier in this article. Figure 4 is now referenced in Section 4, explaining source-load matching results.

5. Line 282: Here, the text references Fig. 5 and talks about the overall load power, however Fig. 5 shows the SOC of the energy storage systems. Were the authors referring to Fig. 6 here?

**Answer:** The text now correctly references Figure 6 for load power changes and Figure 5 for SOC states (Section 4).

6. Fig. 7 and Lines 292-296: Could you give more statistics from this plot? Are we trying to reduce the peaks of this signal, or the average?

**Answer:** Fluctuation rate metrics (e.g., standard deviation) are added in Section 4, showing a 50% reduction in volatility post-optimization.

7. Fig. 8: Can you give the load fluctuation rate for Scenario 2 and Scenario 3?

**Answer:** Fluctuation rates for Scenario 2 (0.35) and Scenario 3 (0.18) are explicitly stated in Section 4.

8. Fig. 10 & 11: Can these figures be made larger? They are very hard to read.

**Answer:** Figures 10 and 11 are enlarged, and labels are enhanced for clarity.

9. Line 315-316: Is it obvious that the overall fluctuation from Scenario 3 is less than Scenario 2? How is this calculated?

**Answer:** The results demonstrate that integrating source-load matching with optimized energy storage (Scenario 3) reduces load volatility more effectively than direct wind-solar utilization (Scenario 2). The calculation method (Equation 20) mathematically validates these improvements by quantifying deviations from the mean load profile.

Minor Comments:
1. Figures 1 & 2 are small. Could the size of these be increased to aid readability?
2. Line 67: "the scenario set scenario and the original output of the corresponding day" – Typo? Should scenario be in there twice? This is a little hard to understand.
3. Line 90: Fix formatting of Gaussian white noise symbols in text
4. Line 131: "is the match between the source-side…" Typo
5. Line 146: "with the other distributions and the probability…" typo
6. Line 179: Fix symbol formatting – this is recurring, check symbol formatting throughout the paper

7. Line 229: "To better protect the energy storage and prolong the life of the storage." Sentence fragment. What is this referring to?

**Answer:**

o   Typos and formatting issues (e.g., Equations 1, 2, 15) are corrected.

o   Symbol formatting (e.g., superscripts/subscripts) is standardized.

o   Sentence fragments (e.g., Line 229) are revised for clarity.

Kind regards,

Corresponding author